# The Perspective of Physical Education Teachers in Spain Regarding Barriers to the Practice of Physical Activity among Immigrant Children and Adolescents: A Qualitative Study

**DOI:** 10.3390/ijerph18115598

**Published:** 2021-05-24

**Authors:** Romain Marconnot, Jorge Pérez-Corrales, Juan Nicolás Cuenca-Zaldívar, Javier Güeita-Rodríguez, Pilar Carrasco-Garrido, Cristina García-Bravo, Eva Solera-Hernández, Sonia Gutiérrez Gómez-Calcerrada, Domingo Palacios-Ceña

**Affiliations:** 1Research Group of Humanities and Qualitative Research in Health Science of Universidad Rey Juan Carlos (Hum&QRinHS), Department of Physical Therapy, Occupational Therapy, Physical Medicine and Rehabilitation, Universidad Rey Juan Carlos, 28922 Alcorcón, Spain; romain.marconnot@urjc.es (R.M.); javier.gueita@urjc.es (J.G.-R.); domingo.palacios@urjc.es (D.P.-C.); 2Rehabilitation Unit, Hospital de Guadarrama, Department of Physical Therapy, Universidad Francisco de Vitoria, 28223 Madrid, Spain; nicolas.cuenca@salud.madrid.org; 3Department of Medical Specialties and Public Health, Universidad Rey Juan Carlos, 28922 Alcorcón, Spain; pilar.carrasco@urjc.es; 4Research Group in Evaluation and Assessment of Capacity, Functionality and Disability of Universidad Rey Juan Carlos (TO+IDI), Department of Physical Therapy, Occupational Therapy, Physical Medicine and Rehabilitation, Universidad Rey Juan Carlos, 28922 Alcorcón, Spain; cristina.bravo@urjc.es; 5Department of Educational Psychology and Psychobiology, Universidad Internacional de La Rioja, Avenida de la Paz 137, 26004 Logroño, Spain; eva.solera@unir.net (E.S.-H.); sonia.gutierrez@unir.net (S.G.G.-C.)

**Keywords:** exercise, physical education and training, school teachers, child, adolescent, emigrants and immigrants, qualitative research

## Abstract

Physical activity (PA) contributes to the development of children and adolescents and to their mental and physical health. The practice of PA in the school context can contribute towards generating a more inclusive educational community for immigrant children and adolescents. The aim of this study was to describe the perspectives of Spanish physical education (PE) teachers on the practice of PA among immigrant children and adolescents. This research was a qualitative exploratory study. A purposeful sampling strategy was used. Data were collected through semi-structured interviews and field notes. Twenty teachers were recruited. An inductive thematic analysis and content analysis were applied. The following topics were identified: (a) Professional expectations; (b) Economic resources; (c) Integration; (d) Family; (e) Religious beliefs and practices; and (f) Gender difference. A predominance of positive emotions was identified in the narratives, and the most repeated words in word clouds were ‘Caribbean’, ‘Latin’, and ‘population’. These findings help to identify PA barriers for immigrant children and adolescents and may contribute to the creation of PA-based interventions in social and educational contexts.

## 1. Introduction

Physical activity (PA) is defined as “*… any bodily movement produced by skeletal muscle that requires energy expenditure*” [1,2]. Physical activity is beneficial at all ages and is recommended as an active practice in early childhood, necessary for the healthy development of both children and adolescents. [3,4,5]. In addition, it prevents diseases such as heart disease, hypertension, or obesity in adulthood [3,4,5]. Furthermore, positive effects on mental health have also been demonstrated, such as a decreased presence of anxious-depressive symptoms in children and adolescents [6,7]. Physical inactivity and sedentary lifestyles have been identified as health-related behaviors that negatively impact quality of life during childhood and youth [2,3,4], noting that worldwide, 80% of adolescents are not sufficiently physically active [8]. Thus, the current recommendation for ages 5 to 17 is an average of 60 min per day of moderate to vigorous PA, mainly aerobic, at least 3 days per week [1,8].

The General Assembly of the United Nations adopted the 2030 Agenda for Sustainable Development Goals, highlighting how PE and opportunities for PA in schools contribute to promoting health and well-being and ensuring quality education in educational communities with different social realities [2,9]. In Europe, the increased immigration of recent decades has led to a large increase in cultural diversity among school-age students in educational contexts [10]. Consequently, educational communities formed by teachers, parents and students are a key element for integration and social cohesion, where the role of PE classes and PE teachers in the generation of inclusive communities for immigrants is highlighted [10]. The practice of PA in the school environment can contribute to improving interpersonal relationships and respect among the different perspectives and cultures that are part of the educational community [11,12], thus fostering feelings of inclusion and belonging to the community [13]. The feelings of inclusion and belonging favors the emotional well-being of immigrant children and adolescents and promotes and facilitates their adaptation to societies they have emigrated to [13].

In Spain, the practice of PA in the school environment has been studied from the perspective of children [14], parents [15,16,17] and teachers [10,15,18,19], both in the general population [14,17] as well as specifically referring to the immigrant population [10,15,16,18,19]. In the study by Martinez-Andrés et al. [14] on the perception of the barriers encountered for the practice of PA in children between 8 and 11 years of age, the opportunities for the practice of PA are determined by the schedules established by their parents, in which the difficulty of reconciling work and family life and the prioritization of other types of academic activities over the practice of PA, together with the perceived danger of the community environment, favor more sedentary habits among the children. These results are consistent with studies carried out from the parents’ perspective [15,16,17], in addition to identifying a greater difficulty for their children to participate in certain physical activities due to economic barriers [15,16,17], a greater difficulty of reconciling single-parent families, restriction of PA among children in informal contexts as a form of punishment, a lack of communication between the school and the family, the influence of the mass media [17], inadequacy of infrastructure and lack of natural environments for practice [15,16,17], excessive homework [15,16] and gender stereotypes and gender-related choices [10,15,16,17,20]. Moreover, Tamura et al. [21] identified that social environments of violence and insecurity in the neighborhoods where children and adolescents live are associated with a decrease in the practice of PA, representing the main structural barrier. These same authors recommend improving the spaces in which PA is carried out, such as parks and other open-air venues. Previous studies [22,23] point to the presence of crime in the neighborhood, the precariousness of the facilities where PA is performed [22,23], and high-traffic levels [23] as barriers to performing PA in children and adolescents in immigrant settings. Recently, Hu et al. [24], in their systematic review on factors influencing the performance of PA in children and adolescents, pointed to unsafe neighborhoods and inaccessibility of PA venues as barriers.

From the teachers’ point of view, different perspectives have been identified, both among PE teachers and among teachers of subjects that take place in the regular classroom (subjects such as mathematics, science, etc.) [18]. For example, the imbalance between previous schooling and the level of the course in which they are enrolled, and the lack of knowledge of the language, are identified as elements that distort children’s learning, according to teachers of classroom-taught subjects, while this issue is not perceived as a problem in the case of physical education (PE) classes [18]. Physical education teachers have been found to have an important role in promoting interculturality in Spanish schools [19]; therefore, the research on their perspective regarding PA practice in this population should be further expanded.

This study was based on the following research questions: How do PE teachers perceive PA in immigrant children and adolescents? What does it mean to them? Answering these questions may help to explain the role of PE teachers and their positioning regarding the promotion of PA practice among immigrant children and adolescents.

The aims were: (a) to describe the perspectives of PE teachers on PA and (b) barriers perceived and (c) to analyze the word frequency from interviews, to assess emotions and (d) to describe the polarity of their perspectives regarding PA.

## 2. Materials and Methods

### 2.1. Design

This study used a qualitative exploratory design [25,26]. Qualitative studies are used to describe people’s experiences and perspectives on programs, complex phenomena involving social context, culture, and education. [25,26]. The design and manuscript structure were inspired by qualitative studies reported by Marconnot et al. [15,16].

The SRQR (Standards for Reporting Qualitative Research) guidelines [27] and the COREQ (Consolidated Criteria for Reporting Qualitative Research) international recommendations were followed [28]. Furthermore, the Guba and Lincoln [25,26] criteria for ensuring the trustworthiness were followed. The various techniques performed to monitor reliability are described in Table 1.

### 2.2. Context

The community of Madrid recorded a foreign population of 15% (1,026,333 inhabitants) in 2020 [29]. Of these, 52.5% were women, 14.15% were under 16 years of age, and 3.48% were between 16 and 19 years of age. The most frequent nationalities among the total foreign population in Madrid were Romania (18.21%), Morocco (8.18%), China (6.41%), Colombia (6.2%), Venezuela (5.91%) and Peru (4.24%). Currently, in Spain, the total number of foreign students is 862,520 (in the community of Madrid 151,603) [30]. In this community, the number of foreign students enrolled in primary education is 59,944 and in secondary education 31,295 [30]. Regarding the number of teachers, in the community of Madrid there are 29,080 teachers in primary education and 21,902 in secondary education [31].

### 2.3. Sampling and Participants

In qualitative studies, sampling is aimed at selecting those participants who have the relevant information about the phenomenon under study (PA practice) [32]. In the present study a non-probabilistic and purposeful sampling strategy were used based on their relevance to the research question rather than representativeness [32]. The inclusion criteria were: (a) PE teachers working in the community of Madrid, (b) who were actively working at an educational center during the study period, and (c) who had over four years’ teaching experience. The exclusion criteria were: (a) teachers who did not teach PE, and (b) not signing the informed consent.

Sampling continued until the ongoing analysis revealed data redundancy [32]. Finally, 20 participants were included and none withdrew from the study.

### 2.4. Data Collection

Semi-structured interviews were used, which followed a question guide oriented to specific areas of interest [25]. The question guide was constructed based on the literature review [15,16] and the experience of the researchers [25]. After obtaining personal and professional data from the participants, the interview began with an opening question: “Please, can you share with me your personal perspective regarding teaching PA to immigrant children and adolescents?” After this, open-ended follow-up questions were used to obtain detailed descriptions: 1. “Could culture or ethnicity influence the acquisition of habits, such as PA?”; 2. “What role do parents play in immigrant children and adolescents’ adherence to PA?”; 3. “While teaching PA, have you experienced any critical event or relevant moment?”; 4. “What do you think are the keys to promoting PA in immigrant children and adolescents?”; 5. “What factors or elements can be considered facilitators or barriers in the acquisition of PA among immigrant children and adolescents?” During the interview, the prompt “Please tell me more about that” was used to encourage participants to explore specific aspects in greater depth.

The interviews were tape-recorded and transcribed verbatim. Overall, 1026 min of data collection were recorded, with a mean of 51.3 min (SD 9.72). All interviews were held at the schools. We used researcher field notes as a secondary source of information to provide more in-depth information [21]. All interviews were conducted in Spanish.

### 2.5. Data Analysis

The interviews were analyzed by means of an inductive thematic analysis [25,32] for the identification of the relevant themes obtained from the interviews, and a content analysis [33] of the participants’ words and narratives. From the content analysis we obtained: (a) a cloud of the most used words, and (b) identification of the feelings of the participants and the polarity of their narratives. The use of content analysis in interviews and written texts through statistical techniques is used in discourse analysis and qualitative studies as a method of deepening and triangulating the analysis [33].

Full transcripts were made of each in-depth interview and of the researchers’ field notes [25,32]. Thematic analysis [25] consisted of identifying text fragments with relevant information to answer the research question. From these narratives, the most descriptive contents (meaning units/codes) were identified. Subsequently, these units were grouped by their common meaning (common meaning groups) and/or similar content [25,32]. During coding, 1049 codes were identified. Thematic analysis was applied separately to interviews and field notes by D.P.-C., J.N.C.-Z. and R.M. Joint team meetings were held to combine the results of the analysis and discuss data collection and analysis procedures. In these team meetings the final themes were displayed, combined, integrated and identified. In case of divergence of opinions, the identification of the theme was based on consensus among the members of the research team. Finally, six themes were identified. See Appendix A Example of codification process.

For the statistical analysis of the qualitative content [33], the software R version 3.5.1 (R Foundation for Statistical Computing, Institute for Statistics and Mathematics, Welthandelsplatz 1, 1020 Vienna, Austria) was used. The text of the interviews was lemmatised for analysis. A word frequency analysis was carried out using the tf-idf algorithm (term frequency-inverse document frequency), and a wordcloud representing the most frequent use of words within the participants’ interviews was obtained. Emotion analysis was performed using Bing [34], Afinn [35] and National Research Council Canada (NRC) dictionaries [36,37,38]. Text polarity was analysed, using the SODictionaries V1.11Spa dictionary as amplifiers and decrementators, the Bing dictionary [36,37,38], and as deniers, those proposed by Vilares et al. [39]. For the analysis of polarity (acceptance-rejection) 4 stages were used. In the first stage, a file was created with the text of the interviews, broken down by sentences for textual analysis. In the second stage, polarity was calculated using the Bing Sentiment Dictionary, amplifiers and de-amplifiers of SODictionar-iesV1.11Spa [36,37,38] and the deniers proposed by Vilares et al. [39] (Appendix A). In the third stage, the scatter diagram of the sentences in the text in relation to neutrality was calculated to identify positive or negative tendencies. Finally, the evolution of emotional valence (positive-negative) was shown throughout the interviews. Fourier transformation was applied to confirm the polarity trend.

### 2.6. Ethics

This study was approved by the Research Ethics Committee of the University Rey Juan Carlos (code: 3001201702417). The study followed the principles of the Declaration of Helsinki). In addition, the present study complied with the Spanish Personal Data Protection Act [40]. All participants provided written informed consent before they participated in the study.

## 3. Results

The study began in September 2019 and ended in September 2020. Twenty participants were included, with a mean age of 40.9 years (SD 6.6) and with 15.55 years (SD 9.2) of PE teaching experience. Of the 20 participants, 13 were male and 7 were female. All participants were currently working in the southern area of the community of Madrid (Spain). Of the participants included, 8 did not believe in any religion, whereas 12 considered themselves to be Catholic Christians. Of the latter, six defined themselves as practising Catholics.

### 3.1. Results of the Thematic Analysis

The themes depict the teachers’ perspective, obtained after analysis of the researchers’ interviews and notes. Six barriers to PA practice were identified: Professional expectations, Economic resources, Integration, Family, Religious beliefs and practices, and Gender difference.

#### 3.1.1. Future Professional Expectations

The participants reported that the parents of immigrant children and youth prioritize other subjects for their children over PE to avoid school failure, especially in the Asian community. Other extracurricular activities, such as English, have more weight than the practice of PA.

The participants spoke of how families with higher levels of education assign higher value to the practice of PA. In addition, from the teachers’ perspective, many families believe that the practice of PA is not necessarily conducive to getting a good job or being successful in society. Moreover, some families reportedly fail to consider the health benefits of PA. In turn, the teachers perceived that certain young people consider PA as a form of leisure, rather than a way of acquiring healthy habits for their lives. In other cases, teachers perceived that some young people view PA as a strategy to avoid family and social conflicts. Thus, PA is seen as a tool that helps young people to overcome problems arising within the family.

#### 3.1.2. Financial Resources

The study participants described that most immigrant families have low or limited financial resources. Furthermore, the teachers pointed out that having low financial resources influences the parent´s ability to purchase clothing, footwear, and sports equipment. Sometimes, the practice of PA implies a substantial economic expense, comprising school or club enrollment fees and the payment of the monthly fee. Teachers reported that this is an expense that many families cannot afford. In addition, in the case of large families, the financial cost of sports equipment is multiplied and becomes a substantial burden for families.
*“Even if the activities are cheap, for an immigrant population that is short of money, spending €10 a month is a lot. If, in addition, there are 5 children, it is €50 a month and that is a lot of money. So, they aren’t going to do it.”* (Teacher#0005#).

Another example of this situation is the performance of optional educational-sports activities at the high school, such as skiing, which immigrant students rarely have access to due to a lack of economic resources. The participants described that, as an alternative, many immigrant students practiced unregulated PA such as playing in the street. Teachers reported that after-school PA programs at the Institutes for the Promotion of Physical Activity and Sport (Institutos Promotores de la Actividad Física y Deporte -IPAFD) promoted with public funds by the community of Madrid are mainly used by immigrant students. These programs cost €14 per year for 2 h of PA per week. This is an expense that families can afford.

#### 3.1.3. Integration

Participants perceived how some communities tend to be quite “close-knitted” and this can lead to difficulties for the integration of children and youth and their ability to practice PA. Indeed, the more integrated the children/youth are within the group and in class, the more they are involved in PA. Moreover, participants narrated how immigrant students tend to form groups among members of the same community, thus an invisible barrier is formed between Spanish students and students from other communities. Some participants used the term “invisible barrier” and this separation was especially identified in young people belonging to the Asian and Maghrebi communities. Some participants mentioned the presence of a kind of “ghetto” phenomenon in PA.

Furthermore, the participants stressed that students who have been in Spain longer, those who are second generation citizens, or who arrived during their childhood display less rejection towards PA. In particular, children and young people from the South American community were highlighted as being able to integrate more easily, because they share a common culture with the Spanish children. In addition, the participants reported how extracurricular activities were identified as a way to promote social integration. Thus, many students enroll in the programs of the IPAFD to integrate better into the class and make friends.

#### 3.1.4. The Family

All participants agreed that the influence of families is key to PA. Thus, the families represent a role model for young people. If the father or mother does not have the habit or possibility of practicing PA, it is difficult for the children to practice a sport.
*“It’s like a vicious circle. Parents don’t do any PA and in the end the children don’t do any PA either. And since the children don’t do anything, the parents don’t practice either.”* (Teacher#0013#).

Another important point stressed by participants was the excessive protection of the families towards girls in relation to boys, when carrying out any type of extracurricular activity. Boys are allowed to move with greater freedom than girls. This perception of the protectionism of families towards girls was observed specifically in the Maghrebi and Asian communities. The participants agreed that the family may perpetuate certain social and cultural stereotypes that place girls at a disadvantage when practicing PA. In addition, the participants pointed out that often the responsibility for the child’s education falls mainly on the mother, due to the physical absence of the father. This means that, in addition to the responsibility of supporting and caring for the rest of the family, the mother relies on the daughters’ help for the performance of household chores or to support in the family business.

#### 3.1.5. Religious Beliefs and Practices

The teachers reported that in some communities, religious practices can hamper or limit the practice of PA for girls, particularly in the Maghrebi community. From the participants ‘perspective, while there are some sports practices that Maghrebi girls can perform with greater or lesser comfort due to the use of other types of clothing, not strictly sportswear, certain sports may cause conflicts with families, such as swimming.

The participants felt that some families may prioritize certain religious practices or religious interpretations over their daughters’ education and learning. This is accentuated during menstruation, as girls begin to wear certain clothes according to their religious beliefs, and in some there is a rejection of PA, sometimes motivated by stereotypes about PA.
*“When they start wearing the veil everything changes. I have seen this change in them in the same class, from being a vital girl, the clothes change, the clothing changes, the temperament changes.”* (#Teacher0010#).
*“…girls are afraid of losing their virginity while practicing gymnastics.”* (Teacher#0001#).

On other occasions, the participants narrated that there are certain practices, such as Ramadan in the Maghrebi community, which involves fasting, and may condition the ability to be physically active during PE classes.

#### 3.1.6. Gender Differences

The teachers reported that there are marked gender roles for the practice of PA. Thus, certain sports are considered to be more suited to boys or girls, especially among the North African and South American communities.
*“They have very much accepted that this is a girl’s thing and that is a boy’s thing.”* (Teacher#0005).

The participants explained that the physical condition of immigrant girls was poorer compared to their Spanish counterparts, in addition to difficulties in hand-eye coordination, rhythm and motor skills. This was especially related to girls from the Maghrebi and Asian communities. The teachers explained the subject of PE in the young girls’ countries of origin was less intense, and that some girls had never developed a habit for practicing sports. In the interviews, the low level of physical fitness only referred to the female population; the participants did not make any comments regarding the boys.

The participants explained that some girls choose to exclude themselves from practicing PA, even during recess or free time, leaving the physical space of the school (playgrounds) to allow the boys to practice PA. The teachers felt that girls perceive themselves as clumsier, or with less skills or abilities than boys. Sometimes girls are even ignored during PE classes or recess when sharing games or sports activities with them.

Some participants believed that the family is not supportive of girls practicing PA, through overprotective attitudes, cultural reasons or religious beliefs.
*“It is a problem of machismo. We can hide it, but it is full-blown machismo, the family can participate and perpetuate certain behaviors with girls.”* (Teacher#0004#).

As a result, many girls are more passive and do not show initiative when it comes to practicing PA; thus, they display signs of apathy or even rejection.
*“Girls tend to be very passive. Physical activity represents a rejection, a refusal to move. For them it seems to be an obligation.”* (Teacher#0013#).

### 3.2. Results of Word Frequency (Wordcloud)

The word cloud revealed that ‘Caribbean’, ‘Latin’ and ‘population’ were the most repeated terms, followed by ‘passive’, ‘Africa’, ‘street’, ‘movement’ and ‘overweight’. See Figure 1.

### 3.3. Results of Emotions and the Polarity (Acceptance-Rejection) Analysis

The sentiment analysis regarding the research questions showed a prevalence of positive emotions (Figure 2A,B). In contrast, according to the Afinn dictionary there is a slight predominance of negative emotions, especially scores of −1, against positive scores of 1 and 2 (Figure 2C). The associated emotions are those of trust and anticipation, followed by fear and joy (Figure 2A).

The interviews presented a polarity of 0.049 ± 0.339, reporting a slight trend to more positive emotions, mainly on account of the presence of extreme positive values (Figure 3) regarding the research questions.

## 4. Discussion

Our findings highlight that, according to the perspective of PE teachers, the educational level of immigrant parents can influence their children’s PA. The survey on the immigrant population in the community of Madrid in 2016 [41] shows that 30% of the immigrant population presented a basic level of education. In Spain, the 2017 national health survey [42] described that the practice of PA was inversely proportional to the families’ level of education. The authors believe that a low level of education could explain a lack of awareness of the benefits of PA on the child’s health and development. The participants in the present study perceived how immigrant families linked PA to leisure and entertainment, giving more importance to other subjects. Along this line, Trigwell et al. [43], show how families from different ethnic communities prioritize educational achievement (based on subjects such as mathematics, science) over PA.

Our results show how family finances can influence the ability to engage in certain physical-sports activities. There are no published data to date on the enrollment of immigrant children and youth in IPAFD programs. The national survey of immigrants in Spain [44] describes that 40% of the immigrant population perform manual or low-skilled jobs, and 22% perform manual jobs that require specific prior training. This emphasizes that many immigrants hold precarious jobs, with low incomes and long working hours. The authors believe that families enter a vicious circle, where the lower the level of education, the less qualified and less well paid the work is, resulting in a lack of sufficient economic resources to enroll their children in extracurricular sports activities, and a decreased practice of PA.

Physical education and sport are important tools in the process of integrating people into society. Lleixà [45] describes how PE is a space for meeting and communication that promotes the relationship between people from different cultures. According to the UNESCO, one of the objectives of PE and sport is to promote proximity between people and individuals [46]. In addition, the International Federation of Physical Education [47] establishes that PE must be a means of combating discrimination and social exclusion. Previous studies [48] show how certain communities may have a more isolated character and less interaction with other communities. Acculturation or assimilation of the host culture of immigrant families is related to the integration of children into society and can be a factor in the acquisition of habits, such as the practice of PA [49,50] Gerber et al. [49] point out that the higher the degree of acculturation of people in a new society, or the longer the stay in the country, the greater the possibilities of practicing PA. Moreover, the lower the level of acculturation in the population, the lower their participation in PA [51,52,53].

Additionally, there are differences between children and adolescents concerning their attitude towards PE lessons. Previous studies report that interest towards PA practice and PE decreases as students get older [54,55,56]. Gómez Rijo et al. [56] describe how the motivation of adolescents in secondary education towards PE practice is lower than that of primary school students. According to Fraile Aranda & Catalina Sancho [57], primary school students tend to feel more skillful at a sporting level than secondary school students. Overall, secondary students have a lower self-concept regarding their ability to perform PA compared to younger students [57].

Among teachers, there are also differences in their perception and orientation towards PA in primary and secondary education influenced by the school curriculum [58,59]. Primary PE teachers orientate learning towards attitudinal and self-realization goals, while secondary school teachers give more importance to procedural and conceptual learning, developing a greater social responsibility [58]. As a result, primary school teachers are more oriented towards games and global developmental work, while secondary school teachers are oriented towards more social skills development. However, Sicilia Camacho et al. [59] point out that in primary education, priority should be given to the development of social relations whereas in secondary education the focus should be on the acquisition of healthy habits.

Another key point is the role of parents as role models for PA practice. Previous studies [60,61] show how parents are an influence on adherence to PA. The more PA practiced by parents in their free time, the more likely their children are to develop similar habits. The authors believe that when parents practice PA, the family has a positive attitude towards PA and the subject of PE, and this can facilitate the practice of PA in their children. However, it should be emphasized that cultural components and/or beliefs can condition PA, particularly in the case of women. This may lead to the existence of a cultural conception of motor activity or motor development in children that is different from that of girls and boys [48,62]. Previous studies [18,63] describe how immigrant girls present a lower level of physical condition and less motor skills when it comes to collective activities, which may result in inhibition and self-exclusion from the sports space.

Most of the participants described religious beliefs as a possible barrier to the practice of PA, in relation to girls from the Maghrebi community. Capllonch et al. [64] describe how certain religious beliefs may limit the practice of PA among female members of the community, aspects such as exposing naked body parts (swimming, athletics), training or sharing PA with males, could prevent girls from participating in sports activities [61,65]. Moreover, previous studies [62] have stressed the need for teacher training on the reality of immigration and especially on the understanding of the cultural aspects of the Maghrebi and Muslim communities. Dagkas et al. [62] in England, describe how PA practice can be adapted for Maghrebi girls while respecting cultural and religious precepts and complying with school rules.

According to our findings, teachers report that girls show less initiative for practicing PA. In Spain, Martínez Baena et al. [66] describe how in students over 13 years of age, 23.1% of the girls practiced PA through obligation, whereas 15.7% acknowledged that they did not practice PA out of laziness. In addition, Flores [48], in a study on multicultural reality and PE teachers, described how immigrant girls claimed to have very little interest and passivity towards the subject of PE.

Regarding differences in teachers’ and parents’ perspectives on PE, Porcuna & Rodríguez-Martín [67] reported that both parents and teachers agree that they can act as role models for children and adolescents, which can positively and negatively influence them to practice PA. Similarly, they agree that there are environmental barriers to PA, such as high workloads, lack of parental time, or the lack of spaces where PA can be carried out [67]. However, in the same study, Porcuna & Rodríguez-Martín [67] described that while teachers demanded greater collaboration between parents and teachers, parents demanded to receive more information from teachers on how to promote PA outside the school context [67]. Similarly, from the parents’ perspective, sex and gender are identified as determining factors in the practice of PA [10,15,16,17,20], along with economic barriers and lack of infrastructure for PA [15,16,17].

It should be noted that of the three dictionaries used to identify the participants’ feelings towards the research questions, two show a positive tendency in their discourse on PA and immigrant children and youth. Only one of the dictionaries used, Afinn, showed negative feelings, but with a percentage very close to positivity. Even though the teachers in the thematic analysis displayed a critical view of the practice of PA in immigrant children, identifying numerous barriers, their discourse tended to be positive, based on the text analysis carried out.

The results of this study reveal the perspective of PE teachers on aspects that may condition the practice of PA in immigrant children and adolescents. This perspective is the object of the study; however, it does not mean that it represents, coincides with or is contrary to the perspective or beliefs of other participants involved in the practice of PA, such as parents and students. This study has several limitations. First, a small number of participants were included. The present study has included 20 participants. Previous qualitative studies [25,30] describe how the total number of participants included does not depend on a previous calculation of the sample size; rather, it is based on the saturation or redundancy of the information obtained in the interviews. Turner-Bowker et al. [68] reported that 92–97% of the saturation can be analyzed after interview number 15 and 20. Secondly, the teachers included in the study were not part of the liaison classrooms, classrooms intended for the support and adaptation of immigrant children/adolescents joining the school. This could influence the results. In addition, teachers were not asked about the process of teaching PE to children and adolescents, which may reveal differences in the performance of PA. Finally, the qualitative nature of this study, gathering the perspective and experiences of a group of PE teachers concerning a certain phenomenon, cannot be generalized. However, it allows us to gain a deeper understanding of the phenomenon from their perspective.

## 5. Conclusions

Our findings help further our understanding of the barriers related to PA in immigrant children and youth in Spain such as giving more importance to certain subjects because of their job prospects, having low economic resources, the lack of role models in the family for the practice of PA, the formation of closed groups for PA practice among students, and the limitation of PA in girls due to gender differences. One of the strengths of this study is that it describes the barriers for performing PA among immigrant children from the perspective of the PE teachers.

Concerning the implications for teaching, these findings may help PE teachers to identify elements such as family issues, beliefs, and economical issues that can condition PA. Furthermore, these results can help young or inexperienced teachers to plan and program the teaching of PE towards the needs of immigrant pupils by promoting greater participation, considering their possible limitations and perspectives. Further, these findings may be considered for the design and implementation of PA programs in certain social contexts. It is important for PA programs to: (a) analyze the academic curriculum and devote increased time to PA, (b) provide financial support to families who have limited resources in order to cover any associated costs related to children’s sports practice, and (c) promote the organization of sports activities and multicultural games in community settings.

Future lines of research should include the perspective of other participants, such as students and parents, regarding PA. In addition, it is important to examine the barriers for practicing PA in different contexts (school environment, neighborhood), and identify any obstacles for the practice of PA among immigrant students of different nationalities, also from a gender perspective.

## Figures and Tables

**Figure 1 ijerph-18-05598-f001:**
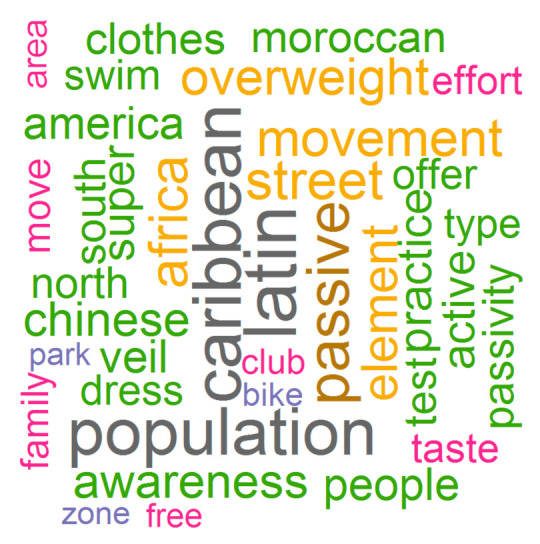
Word cloud from interview responses after applying the tf-idf algorithm.

**Figure 2 ijerph-18-05598-f002:**
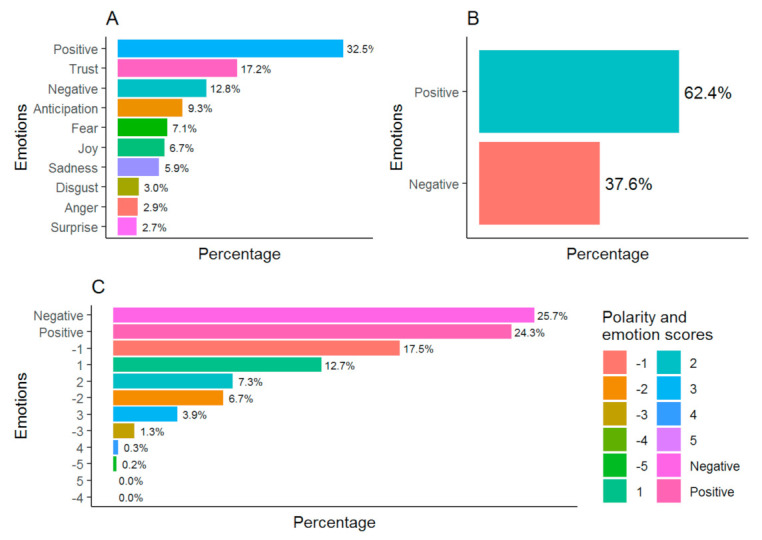
TNRC (**A**), Bing (**B**) and Afinn (**C**) dictionaries sentiment scores.

**Figure 3 ijerph-18-05598-f003:**
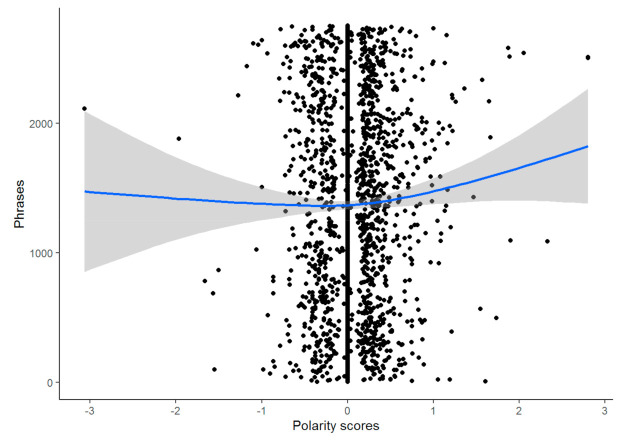
Polarity by phrases.

**Table 1 ijerph-18-05598-t001:** Trustworthiness criteria.

Criteria	Techniques Performed and Application Procedures
Credibility	Investigator triangulation: Team meetings were organised during the thematic analysis, the results were compared, and the final results were identified.
Transferability	In-depth descriptions of the study, providing data and describing the study design and its different sections (context, research team, reflexivity process, sampling, inclusion criteria, data collection, and analysis).
Dependability	Audit by an external researcher, responsible for the assessment of the study protocol, with a special focus on the method and process of implementation during the study.
Confirmability	Investigator triangulation, data collection and analysis triangulation (thematic and content analysis).The process of reflexivity was conducted through the description of the researchers’ positioning; reflective debriefing by the researchers during data collection and analysis.

## Data Availability

Personal data are stored in the data protection file of the Universidad Rey Juan Carlos. This is a qualitative research we could not provide transcribed files, we followed Spanish Organic Law on Protection of Personal Data and guarantee of digital rights (2018).

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
