# Peer review of "The Perspective of Physical Education Teachers in Spain Regarding Barriers to the Practice of Physical Activity among Immigrant Children and Adolescents: A Qualitative Study"

_ijerph, 2021, doi:10.3390/ijerph18115598_

Round 1
Reviewer 1 Report
The subject of the study is interesting, but the objective of the study does not really analyze the physical activity barriers of the students, but rather the teachers' perception of these barriers. In fact, it only lists and describes the perspectives of a group of physical education teachers on the practice of physical activity among immigrant children.
And in some cases the interpretation goes beyond the evidence. For example, in the paragraph: “Study participants described that most immigrant families have scarce or limited financial resources. This influences your ability to buy sports clothing, footwear, and equipment. Thus, the practice of PA implies a substantial economic expense, which includes school or club tuition and the payment of the monthly fee. This is an expense that many families cannot afford. In addition, in the case of large families, the economic cost of sports equipment multiplies and becomes a substantial burden for families.
If families have financial resources, why is it said that "This influences their ability to purchase clothing, footwear and sports equipment" and that, therefore, the practice of PA implies a substantial economic expense, which includes school or club tuition and the payment of the monthly fee. This is an expense that many families cannot afford.
The number of participants is really small (n = 20), although it is noted in the article that the sampling is continued until the ongoing analysis revealed data redundancy.
The inclusion criteria were physical education teachers working in the Community of Madrid. It would be important to know how many teachers teach in the Community of Madrid. And how many of them work with immigrant children since this criterion does not appear as an inclusion criterion. Given the size of the Madrid community, it is difficult to ensure that the information is already significant with 20 participants. That would mean that almost everyone thinks the same. What is difficult in these topics.
By increasing the sample or triangulating populations, it would be possible to access information with a higher level of divergence.
It would be necessary to specify in detail how the analysis categories were extracted since it only indicates "To analyze the teachers' experience an inductive thematic analysis was performed 165 on the collected data"
The statistical analysis is well designed. But the conclusions are really poor.
For example, in “Professional expectations” do the participants who are teachers think about what the parents of the students say? It is not direct information. Have teachers asked parents? Populations could be triangulated and parents and students could also be interviewed in order to make these claims.
Teachers give their opinion on the financial resources of families or their emotions. It would not be more logical then to ask the families directly.
In addition, it would be important to add that implications of the study are derived for teaching practice if the barriers to the practice of physical activity had actually been analyzed.
Definitely:
1.- Adapt the title of the article to the objective. Physical activity barriers are not really studied, but the teachers' perception of these barriers.
2.- The introduction should describe the background on the research on barriers to physical activity in immigrants
3.- Write the objective in a manner consistent with the title
4.- Increase the sample size or triangulate the population with interviews with families and students
5.- Detail the quantitative data by which the results are reached in each of the categories.
6.- The results must be the result of the analysis of the interviews and not of the authors' conjectures
7.- Describe the educational implications of these findings
Author Response
REVIEWER 1
Comments and Suggestions for Authors
The subject of the study is interesting, but the objective of the study does not really analyze the physical activity barriers of the students, but rather the teachers' perception of these barriers. In fact, it only lists and describes the perspectives of a group of physical education teachers on the practice of physical activity among immigrant children.
Response: To determine the objective of the study, this was based on the following study question: “The questions guiding this study were: How do physical education teachers perceive PA in immigrant children and adolescents?” The research question directing the study already includes the fact that the aim of this study will be to describe the teachers' perspective on physical activity (phenomenon under study). This way of constructing the question and directing the research objective is used in qualitative research.
We do not see the fact that we describe the perspective of a group of people as a negative aspect or a limitation, but rather as part of the design of qualitative research studies aimed at describing and analysing the perspective of groups living in or in contact with certain phenomena.
Carpenter & Suto (2008) reported that: “…qualitative research is well suited to answer questions that ask what particular experiences are like and how people create meaning from their circumstances. These types of questions are appropriate for topics where Little is known from the client´s point of view, and much depends on how knowledge is created…” (page 21).
However, based on the reviewer's comments, the objectives of the study have been modified to clarify them:
The aims were: a) to describe the perspectives of PE teachers on PA and b) barriers perceived, and c) to analyze the word frequency from interviews, to assess emotions and d) to describe the polarity of their perspective regarding PA.
References
- Carpenter C, Suto M. Qualitative Research For Occupational And Physical Therapist. Oxford, UK: Blackwell Publishing; 2008.
And in some cases the interpretation goes beyond the evidence. For example, in the paragraph: “Study participants described that most immigrant families have scarce or limited financial resources. This influences your ability to buy sports clothing, footwear, and equipment. Thus, the practice of PA implies a substantial economic expense, which includes school or club tuition and the payment of the monthly fee. This is an expense that many families cannot afford. In addition, in the case of large families, the economic cost of sports equipment multiplies and becomes a substantial burden for families.”
If families have financial resources, why is it said that "This influences their ability to purchase clothing, footwear and sports equipment" and that, therefore, the practice of PA implies a substantial economic expense, which includes school or club tuition and the payment of the monthly fee. This is an expense that many families cannot afford.
Response: The authors confirm that the results are based on the participants' narratives, and that they have been obtained after the coding process during the inductive thematic analysis. In this study, two types of analysis of the narrative material have been carried out: an inductive thematic analysis, where the identified themes are described, and, on the other hand, a textual content analysis. In this manner, the results section shows different types of results obtained by different analysis proposals. The results of the thematic analysis appear in the section: 3.1. Results of the thematic analysis. 3.2. Results of word frequency (wordcloud), and 3.3. Results of emotions and the acceptance-rejection (polarity) analysis.
The results of the thematic analysis are derived from the identification of codes and their subsequent classification and grouping without the use of statistical calculations. This identification and classification (coding process) is based on the participants' narratives (derived from the interviews) that present relevant information.
Moreover, the authors have included at results section the narratives and extracts from the participants because the international recommendations and the consensus on quality in qualitative health sciences studies, such as the Consolidated criteria for reporting qualitative research (COREQ) (Tong et al., 2007) and the Standards for reporting qualitative research (SRQR) (O'Brien et al., 2014), recommend that in the results section, real examples should be used from the participants’ narratives that justify the analysis, description of the researchers.
“The presentation of results often varies with the specific qualitative approach and methodology; thus, rigid rules for reporting qualitative findings are inappropriate. However, authors should provide evidence (e.g., examples, quotes, or text excerpts) to substantiate the main analytic findings.” (O'Brien et al., 2014.p.1247-1248)
Tong et al (2007) reported at Domain 3: analysis and findings at COREQ guidelines (Tong et al., 2007): “The credibility of the findings can be assessed if the process of coding (selecting significant sections from participant statements), and the derivation and identification of themes are made explicit. (p.356).
This justification and proof of the results, via the inclusion of real examples and narratives of the participants is called credibility of the results (Cohen & Crabtree, 2008; Shenton 2004; Tong et al., 2007).
On the other hand, in order to study the perspective of the participants included, it is not necessary to use questionnaires or quantification of narratives (Curry et al., 2009; Creswell & Poth, 2018; Carpenter & Suto, 2008; Pope & Mays, 2006). Qualitative research does not mean that it is a type of research that uses qualitative variables that are quantifiable and analyzed (Curry et al., 2009; Creswell & Poth, 2018). Essentially, qualitative research mainly works with qualitative data such as transcribed narrative texts (obtained in the interviews), images (pictures, photographs), and written documents (letters, diaries).
“Qualitative data can be derived from interaction with participants, such as interviews or focus groups, or as a result of participant observation or analysis of documents or records. Qualitative data, whether in the form of transcripts or field notes, are generally presented in narrative form. When derived from interaction with participants the data are presented as verbatim quotations, to preserve and represent the voice of the participants.” (Carpenter & Suto, 2008, p.32)
Qualitative research is a type of research that is based on studying the experience and perspective of people in certain situations and how they experience the impact of interventions and programms (Curry et al., 2009; Creswell & Poth, 2018; Carpenter & Suto, 2008; Pope & Mays, 2006). In the PubMed database, in their Tesaurus section, “qualitative research” is defined as: “Any type of research that employs nonnumeric information to explore individual or group characteristics, producing findings not arrived at by statistical procedures or other quantitative means.”
Creswell & Poth (2018) defined qualitative research as: “Qualitative research begins with assumptions and the use of interpretive/theoretical frameworks that inform the study of research problems addressing the meaning individuals or groups ascribe to a social or human problem. To study this problem, qualitative researchers use an emerging qualitative approach to inquiry, the collection of data in a natural setting sensitive to the people and places under study, and data analysis that is both inductive and deductive and establishes patterns or themes. The final written report or presentation includes the voices of participants, the reflexivity of the researcher, a complex description and interpretation of the problem, and its contribution to the literatura or a call for change.”(p.8)
References:
- Carpenter C, Suto M. Qualitative research for occupational and physical therapists: A practical guide. Oxford: Black-Well Publishing, 2008.
- Creswell JW, Poth CN. Qualitative inquiry and research design. Choosing among five approaches. 4 ed. Thousand Oaks: SAGE, 2018.
- Cohen DJ, Crabtree BF. Evaluative criteria for qualitative research in health care: controversies and recommendations. Ann Fam Med 2008;6:331–9.
- Curry LA, Nembhard IM, Bradley EH. Qualitative and Mixed Methods Provide Unique Contributions to Outcomes Research. Circulation. 2009; 119: 1442–1452.
- Curry L, Nuñez-Smith M. Mixed methods in health sciences research. Thousand Oaks, CA: Sage publications; 2015.
- Herbert R, Jamtvedt G, Hagen KB, Mead J. Practical Evidence-Based Physiotherapy. 2 ed. Elsevier Churchill Livingstone: Oxford, England, 2011.p. 28-29.
- O'Brien BC, Harris IB, Beckman TJ, Reed DA, Cook DA. Standards for reporting qualitative research: a synthesis of recommendations. Acad Med. 2014;89(9):1245-1251.
- Pope C, Mays N. Qualitative research in health care. Oxford: Blackwell Publishing; 2006.
- Tong A, Sainsbury P, Craig J. Consolidated criteria for reporting qualitative research (COREQ): a 32-item checklist for interviews and focus groups. Int J Qual Health Care. 2007;19(6):349–57.
- Shenton AK. Strategies for ensuring trustworthiness in qualitative research projects. Education for Information. 2004;22:63–75.
However, the authors believe that the wording of the text of the thematic results may have been misleading making it appear to be the researchers' assertions rather than those of the participants. Therefore, the wording of this paragraph and other paragraphs of the results section have been modified to make it clear that we describe the perspective of the teachers, not the opinion of the researchers.
The following text has been inserted:
The participants spoke of how families with higher levels of education assign higher value to the practice of PA. Also, from the teachers´ perspective, many families believe that the practice of PA is not necessarily conducive to getting a good job or being successful in society.
The study participants described that most immigrant families have low or limited financial resources. Furthermore, the teachers pointed out that having low financial resources influences the parent´s ability to purchase clothing, footwear, and sports equipment. Sometimes, the practice of PA implies a substantial economic expense, comprising school or club enrollment fees and the payment of the monthly fee. Teachers reported that this is an expense that many families cannot afford. In addition, in the case of large families, the financial cost of sports equipment is multiplied and becomes a substantial burden for families.
Another example of this situation is the performance of optional educational-sports activities at the high school, such as skiing, which immigrant students rarely have access to due to the lack of economic resources. The participants described that, as an alternative, many immigrant students practiced unregulated PA such as playing in the street.
Moreover, participants narrated how immigrant students tend to form groups among members of the same community, thus an invisible barrier is formed between Spanish students and students from other communities. Some participants used the term “invisible barrier” and this separation was especially identified in young people belonging to the Asian and Maghrebi communities.
In addition, the participants reported how extracurricular activities were identified as a way to promote social integration.
Another important point stressed by participants was the excessive protection of the families towards girls in relation to boys, when carrying out any type of extracurricular activity.
The teachers reported that in some communities, religious practices can hamper or limit the practice of PA for girls, particularly in the Maghrebi community. From the participants‘ perspective, while there are some sports practices that Maghrebi girls can perform with greater or lesser comfort due to the use of other types of clothing, not strictly sportswear, certain sports may cause conflicts with families, such as swimming.
On other occasions, the participants narrated that there are certain practices, such as Ramadan in the Maghrebi community, which involves fasting, and may condition the ability to be physically active during PE classes.
On the other hand, the information that was included is that families had limited or low resources. “Study participants described that most immigrant families have scarce or limited financial resources.” For this reason, from the teachers' perspective, families were not going to be able to buy sports equipment or participate in sports registrations. The authors believe that the wording and information provided is correct.
The number of participants is really small (n = 20), although it is noted in the article that the sampling is continued until the ongoing analysis revealed data redundancy.
Response: We thank the reviewer for the first comments, however, regarding the last observation, the authors de not agree with the statement that the small number of patients of the same undermine the value of this study.
Regarding the number of patients, the reasons are as follows; in qualitative research there is not a number of participants that one can calculate previously (Creswell & Poth, 2018; Carpenter & Suto, 2008), in the same manner, in qualitative research we do not seek to extrapolate the results to the general population (Creswell & Poth, 2018; Pope & Mays, 2006), rather, the attempt is to deepen our knowledge of the complex phenomena (physical activity promotion) or impact of programs and interventions on groups of people who live specific situations.
“The intent in qualitative research is not to generalize the information but to elucidate the particular, the specific.” (Creswell & Poth, 2018.p.158).
“In quantitative studies, power calculations determines which sample size (N) is necessary to demonstrate effects of a certain magnitude from an intervention. For qualitative interview studies, no similar standards for assessment of sample size exist.” (Malterud et al., 2015.p.1)
“Qualitative researchers actively seek to recruit participants who can represent well and have experience of the phenomenon of interest (…)The key is to focus the participant selection strategically and meaningfully, rather than to create a representative sample (…) The sample size and range of participants are intended to generate sufficient data to explore processes, similarities and differences, to develop theory and descriptions that take into account specific contexts, rather than to make statistical comparisons or infer causal relationships. In qualitative research there is no set formula for determining the sample size (…) The quality and comprehensiveness of the data acquired varies with different participants or sources of information. Some participants have more experiences with the topic of interest, are able to reflect deeply on their experiences (…) The greater the quality of data acquired from each source, the smaller the sample size required.” (Carpenter & Suto, 2008.p.80).
Besides, the inclusion of a greater or lesser number of participants depends on the richness of the information that is obtained and the presence of repetition of this information among the participants who are included (Creswell & Poth, 2018; Carpenter & Suto, 2008; Malterud et al., 2015). In qualitative research, this concept is known as “data saturation” or “information redundancy” (Malterud et al., 2015). In this manner, in this study, 20 participants were included because in this number data redundancy was obtained (saturation) (Creswell & Poth, 2018; Carpenter & Suto, 2008; Malterud et al., 2015) this means that the contents of the interviews analyzed present repetitive narratives or results, therefore it is not necessary to continue with the data collection and/or the inclusion of more participants (Creswell & Poth, 2018; Carpenter & Suto, 2008). Recently, Turner-Bowker et al. (2018), reported that saturation or redundant information is described as the point at which no new data is expected to emerge from the performance of additional qualitative interviews. These authors in their studies showed to assess the point at which saturation of concept was achieved in a retrospective evaluation of 26 interview studies between 2006 and 2013. The findings reported that 92% of redundant information emerged by the 15th interview and 97% emerged by the 20th interview. For this reason, the authors of this study believe that are number of participants (n=20) is optimal and well suited to the saturation of data obtained after the analysis of our interviews following the qualitative methodology and the recommendations for qualitative studies in health sciences established by the Consolidated criteria for reporting qualitative research (COREQ) (Tong et al., 2007) and the Standards for reporting qualitative research (SRQR) (O'Brien et al., 2014).
However, following the reviewer’s recommendations, we believe it is necessary to include more information in the manuscript (limitations section) so that this is clearer.
We have included this text in the ‘limitations’ section:
This study has several limitations. First, a small number of participants was included. The present study has included 20 participants. Previous qualitative studies [25,30] describe how the total number of participants included does not depend on a previous calculation of the sample size, rather it is based on the saturation or redundancy of the information obtained in the interviews. Turner-Bowker et al. [68] reported that 92-97% of the saturation can be analyzed after interview number 15 and 20.
References:
- Carpenter C, Suto M. Qualitative research for occupational and physical therapists: A practical guide. Oxford: Black-Well Publishing, 2008.
- Creswell JW, Poth CN. Qualitative inquiry and research design. Choosing among five approaches. 4 ed. Thousand Oaks: SAGE, 2018.
- Malterud K, Siersma VD, Guassora AD. Sample Size in Qualitative Interview Studies: Guided by Information Power. Qual Health Res. 2015 Nov 27. DOI: 10.1177/1049732315617444
- O'Brien BC, Harris IB, Beckman TJ, Reed DA, Cook DA. Standards for reporting qualitative research: a synthesis of recommendations. Acad Med. 2014;89(9):1245-1251.
- Pope C, Mays N. Qualitative research in health care. Oxford: BMJ Books & Blackwell Publishing; 2006.
- Tong A, Sainsbury P, Craig J. Consolidated criteria for reporting qualitative research (COREQ): a 32-item checklist for interviews and focus groups. Int J Qual Health Care. 2007;19(6):349–57.
- Turner-Bowker DM, Lamoureux RE, Stokes J, Litcher-Kelly L, Galipeau N, Yaworsky A, Solomon J, Shields AL. Informing a priori Sample Size Estimation in Qualitative Concept Elicitation Interview Studies for Clinical Outcome Assessment Instrument Development. Value Health. 2018 Jul;21(7):839-842. doi: 10.1016/j.jval.2017.11.014.
We have included a new reference in the manuscript:
- Turner-Bowker DM, Lamoureux RE, Stokes J, Litcher-Kelly L, Galipeau N, Yaworsky A, et al. Informing a priori Sample Size Estimation in Qualitative Concept Elicitation Interview Studies for Clinical Outcome Assessment Instrument Development. Value Health 2018;21:839–42. PubMed https://doi.org/10.1016/j.jval.2017.11.014
The inclusion criteria were physical education teachers working in the Community of Madrid. It would be important to know how many teachers teach in the Community of Madrid. And how many of them work with immigrant children since this criterion does not appear as an inclusion criterion. Given the size of the Madrid community, it is difficult to ensure that the information is already significant with 20 participants. That would mean that almost everyone thinks the same. What is difficult in these topics.
Response: As indicated in the previous question on sample size, qualitative studies do not seek for the results to be representative, nor for these to be extrapolated to the whole population. The aim is to gain an in-depth understanding of groups of people who are subject to certain situations, who experience complex phenomena or who live in different contexts. The justification for sample size and qualitative studies has been developed in the previous question.
On the other hand, from the perspective of the researchers, it is not possible to use as a criterion for inclusion the fact that we will only choose teachers who work with immigrant children and adolescents, because there are no schools or high schools in the Community of Madrid that only cater for immigrants. The authors believe that if there were such schools/institutes in the public education system, this would lead to a "segregation" of students, with important legal and ethical repercussions. Because of this, in the present study, we included teachers who currently had immigrant children and adolescents in their physical education classes.
Moreover, it is true that there are liaison classes, created to support immigrant children and adolescents entering the Spanish education system. In these classrooms, specific work is carried out on this point, and this could influence the results of the participants.
The authors agree with the reviewer that it is necessary to report on this point. Therefore, it has now been included in the limitations:
Secondly, the teachers included in the study were not part of the liaison classrooms, classrooms intended for the support and adaptation of immigrant children/adolescents joining the school. This could influence the results.
The authors have not found any documentation indicating the ratio of immigrant teachers/students in the Community of Madrid in the databases of the Spanish Ministry of Education and Vocational Training (https://www.educacionyfp.gob.es/portada.html).
However, the authors believe that the reviewer is right and it would be necessary to include information on the number of teachers in primary and secondary education in the Community of Madrid, and the number of immigrant students in primary and/or secondary education.
We have included the following text in the context section:
The community of Madrid recorded a foreign population of 15% (1026,333 inhabitants) in 2020 [29]. Of these, 52.5% were women, 14.15% were under 16 years of age, and 3.48% were between 16 and 19 years of age. The most frequent nationalities among the total foreign population in Madrid were Rumania (18.21%), Morocco (8.18%), China (6.41%), Colombia (6.2%), Venezuela (5.91%) and Peru (4.24%). Currently, in Spain, the total number of foreign students is 862,520 (in the Community of Madrid 151,603) [30]. In this Community, the number of foreign students enrolled in primary education is 59,944 and in secondary education 31,295 [30]. Regarding the number of teachers, in the Community of Madrid there are 29,080 teachers in primary education and 21,902 in secondary education [31].
We have included new references in the manuscript:
- Ministerio de Educación y Formación Profesional [Ministry of Education and Vocational Training]. Enseñanzas no universitarias. Alumnado matriculado. Curso 2019-2020. Resultados detallados. Alumnado extranjero. Alumnado extranjero por titularidad/financiación del centro, comunidad/provincia y enseñanza. Ministerio de Universidades [Ministry of Universities]: Madrid, 2021. Available online: http://estadisticas.mecd.gob.es/EducaDynPx/educabase/index.htm?type=pcaxis&path=/no-universitaria/alumnado/matriculado/2019-2020-rd/extran&file=pcaxis&l=s0 (accessed on 12 may 2021).
- Ministerio de Educación y Formación Profesional [Ministry of Education and Vocational Training]. Enseñanzas no universitarias. Estadística del profesorado y otro personal. Curso 2019-2020. Resultados detallados. Profesorado por comunidad autónoma/provincia, titularidad y dedicación. Ministerio de Universidades [Ministry of Universities]: Madrid, 2021. Available online: http://estadisticas.mecd.gob.es/EducaJaxiPx/Tabla.htm?path=/no-universitaria/profesorado/estadistica/2019-2020-rd/reggen//l0/&file=reggen06.px&type=pcaxis&L=0 (accessed on 12 may 2021).
By increasing the sample or triangulating populations, it would be possible to access information with a higher level of divergence.
Response: The authors believe that the solution given by the reviewer could be applied to population and ecological studies, based on quantitative epidemiological designs such as cross-sectional studies, cohort studies, observational studies, where sample calculation is essential for external validity (Elmore et al., 2020).
However, in the case of qualitative studies, the aim is not to study populations, neither through the description and analysis of a phenomenon or the perspective of a group of people (teachers) for the purpose of extrapolating these results to the rest of the population. Qualitative research is used to understand, not to quantify or extrapolate.
As we have responded earlier, in qualitative research, there is not a number of participants that one can calculate previously (Creswell & Poth, 2018; Carpenter & Suto, 2008), in this same manner, in qualitative research we do not seek to extrapolate the results to the general population (Creswell & Poth, 2018; Pope & Mays, 2006), but rather, the intention is to deepen our knowledge of the impact of programs and interventions in groups of people who suffer from illnesses or specific situations of health and illness.
“The intent in qualitative research is not to generalize the information but to elucidate the particular, the specific.” (Creswell & Poth, 2018.p.158).
“In quantitative studies, power calculations determines which sample size (N) is necessary to demonstrate effects of a certain magnitude from an intervention. For qualitative interview studies, no similar standards for assessment of sample size exist.” (Malterud et al., 2015.p.1)
“Qualitative researchers actively seek to recruit participants who can represent well and have experience of the phenomenon of interest (…)The key is to focus the participant selection strategically and meaningfully, rather than to create a representative sample (…) The sample size and range of participants are intended to generate sufficient data to explore processes, similarities and differences, to develop theory and descriptions that take into account specific contexts, rather than to make statistical comparisons or infer causal relationships. In qualitative research there is no set formula for determining the sample size (…) The quality and comprehensiveness of the data acquired varies with different participants or sources of information. Some participants have more experiences with the topic of interest, are able to reflect deeply on their experiences (…) The greater the quality of data acquired from each source, the smaller the sample size required.” (Carpenter & Suto, 2008.p.80).
Qualitative designs are among the different types of research that can be used to describe small-scale, in-depth phenomena. They are thus featured in the initiative EQUATOR- Enhancing the QUAlity and Transparency Of health Research (https://www.equator-network.org/). Also worth bearing in mind are the qualitative methodology and the recommendations for qualitative studies established by the Consolidated criteria for reporting qualitative research (COREQ) (Tong et al., 2007), the Standards for reporting qualitative research (SRQR) (O'Brien et al., 2014), and The APA Publications and Communications Board task force for reporting standards for qualitative primary, qualitative meta-analytic, and mixed methods research (Levitt et al., 2018).
References:
- Elmore J, Wild D, Nelson H, Katz D. Jekel's Epidemiology, Biostatistics, Preventive Medicine, and Public Health. 5th Edition. Elsevier, USA: 2020.
- Levitt HM, Bamberg M, Creswell JW, Frost DM, Josselson R, Suárez-Orozco C. Journal article reporting standards for qualitative primary, qualitative meta-analytic, and mixed methods research in psychology: The APA Publications and Communications Board task force report. Am Psychol. 2018 Jan;73(1):26-46. doi: 10.1037/amp0000151. PMID: 29345485.
- O'Brien BC, Harris IB, Beckman TJ, Reed DA, Cook DA. Standards for reporting qualitative research: a synthesis of recommendations. Acad Med. 2014;89(9):1245-1251.
- Tong A, Sainsbury P, Craig J. Consolidated criteria for reporting qualitative research (COREQ): a 32-item checklist for interviews and focus groups. Int J Qual Health Care. 2007;19(6):349–57.
It would be necessary to specify in detail how the analysis categories were extracted since it only indicates "To analyze the teachers' experience an inductive thematic analysis was performed on the collected data".
Response: The authors have included further information on inductive thematic analysis. In addition, a supplementary file (Supplementary material 1) "example of codification process" is included, which graphically describes the codification process from the narratives and shows an example of its development.
We have included new information regarding inductive thematic analysis:
The interviews were analyzed by means of an inductive thematic analysis [25, 32] for the identification of the relevant themes obtained from the interviews, and a content analysis [33] of the participants' words and narratives. From the content analysis we obtained: a) a cloud of the most used words, b) to identify the feelings of the participants and the polarity of their narratives. The use of content analysis in interviews and written texts through statistical techniques is used in discourse analysis and qualitative studies as a method of deepening and triangulating the analysis [33].
Full transcripts were made of each in-depth interview and of the researchers' field notes [25, 32]. Thematic analysis [25] consisted of identifying text fragments with relevant information to answer the research question. From these narratives, the most descriptive contents (meaning units/codes) were identified. Subsequently, these units were grouped by their common meaning (common meaning groups) and/or similar content [25, 32]. During coding, 1049 codes were identified. Thematic analysis was applied separately to interviews and field notes by DPC, JNCZ and RM. Joint team meetings were held to combine the results of the analysis and discuss data collection and analysis procedures. In these team meetings the final themes were displayed, combined, integrated and identified. In case of divergence of opinions, the identification of the theme was based on consensus among the members of the research team. Finally, 6 themes were identified. See Supplementary file, Figure S1 Example of codification process.
We have included more information regarding the codification process - see supplementary file 1. Figure S1 Example of codification process.
The statistical analysis is well designed. But the conclusions are really poor.
For example, in “Professional expectations” do the participants who are teachers think about what the parents of the students say? It is not direct information. Have teachers asked parents? Populations could be triangulated and parents and students could also be interviewed in order to make these claims. Teachers give their opinion on the financial resources of families or their emotions. It would not be more logical then to ask the families directly.
Response: We have divided the response into 3 parts based on the reviewer's comments.
Regarding statistical analysis and poor conclusions. (The statistical analysis is well designed. But the conclusions are really poor. For example, in “Professional expectations”)
The researchers in this qualitative study conducted two types of analysis. On the one hand, an inductive thematic analysis, which focuses on identifying relevant fragments of the text, based on the relevant information they provide to answer the research question and its objectives. Thematic analysis occurs through the coding process, where the most relevant categories are identified, based on the information they provide (quality), and by consensus among researchers, not their frequency (Vaismoradi et al., 2014).
Furthermore, a second analysis of the qualitative material obtained from the interviews was carried out, a content analysis (Vaismoradi et al., 2014), based on the quantification of the most frequent words and through the use of dictionaries of standardised terms to obtain the polarity of the discourse/narrative of the teachers and the emotions in their discourse. This analysis has involved statistical treatment of the data.
The reviewer notes that the statistical treatment is well designed, but that the results are poor. One of the results from the inductive thematic analysis, which appears in the manuscript as part of the section, is pointed out as an example:
3.1. Results of the thematic analysis.
The results presented in this section derive from the thematic analysis, not from the content analysis, where the statistical treatment has been carried out. Therefore, the authors do NOT believe that the results of the thematic analysis (as indicated in this example) are poor, but rather that they come from a coding process, based on the relevance of the information and subsequent consensus among researchers. It does not come from a statistical treatment of the data.
The results of the content analysis and statistical processing are presented in the results presented in the following sections of the manuscript:
3.2. Results of word frequency (wordcloud)
3.3. Results of emotions and the acceptance-rejection (polarity) analysis
Therefore, the authors believe that the results presented are methodologically well-founded, based on the process of analysis carried out in each of them, and described and justified in the methods section.
References:
- Vaismoradi M, Turunen H, Bondas T. Content analysis and thematic analysis: Implications for conducting a qualitative descriptive study. Nurs Health Sci. 2013 Sep;15(3):398-405. doi: 10.1111/nhs.12048. Epub 2013 Mar 11. PMID: 23480423.
Regarding triangulation and include or not more participants groups. (Have teachers asked parents? Populations could be triangulated and parents and students could also be interviewed in order to make these claims)
Response:
The authors agree with the reviewer that triangulation could be applied to a) confirm information, b) contextualise findings, and c) deepen findings (Creswell & Poth, 2018).
However, the authors believe that it is not possible to include a new phase of data collection in the present study, or to include new participants, once the study has been completed and finalised. For the following reasons: a) the present study has been created following a specific research question, to get to know the teachers' perspective, focusing on defined participants. It is not possible to include elements not covered by the research question a posteriori; b) including new participants and new data collection would mean creating and starting a new, different study from scratch; and c) as with any study, the project should be submitted before the study is conducted for evaluation by an ethical research committee to determine the ethical and legal considerations in conducting studies on minors (specially protected population).
What has been done in the present study, to ensure the credibility of the results (following Lincoln and Guba's criteria) is to apply a process of triangulation between researchers (Creswell & Poth, 2018), in the thematic analysis, by means of 3 researchers (DPC, JNCZ, and RM) to the material obtained by the participants. We have now included this information in the analysis section.
We have included the following text:
Thematic analysis was applied separately to interviews and field notes by DPC, JNCZ and RM. Joint team meetings were held to combine the results of the analysis and discuss data collection and analysis procedures.
Direct or indirect information and participants´ perspective construction. (do the participants who are teachers think about what the parents of the students say? It is not direct information. Have teachers asked parents? Teachers give their opinion on the financial resources of families or their emotions. It would not be more logical then to ask the families directly.)
Response: We do not understand this observation. In our view, it depends on the research question (Creswell & Poth, 2018). If the research question includes other participants such as parents, then this information should be collected. However, if they are not included in the question, then it should not be collected.
In this study, the guiding research question is the teachers' perspective on physical activity in immigrant children and what elements, from their perspective (Creswell & Poth, 2018), are relevant and may influence (positively or negatively) promoting and engaging in physical activity. Teachers narrate their individual perspective on that phenomenon. This implies that from their perspective, more 'social actors' or elements such as parents, spiritual leaders, the presence of infrastructure, insecurity on the streets, etc. could be involved. Qualitative designs do not attempt to control the perspective of their participants, but to describe their experience in their context, under the conditions they live in, and always from their perspective (Creswell & Poth, 2018).
Therefore, it seems justified and coherent to the authors to ask teachers about their perspective on physical activity and immigrant children and adolescents. Moreover, one cannot guarantee that elements such as economic issues, or other issues that may influence their children's physical activity, will not appear in the parents' interviews.
Subjectivity is part of qualitative designs, which is why qualitative research does not seek to be representative, but rather to understand a specific phenomenon (physical activity) in depth, among specific people (physical education teachers), living in a specific context (school environment) (Creswell & Poth, 2018; O'Brien et al., 2014; Tong et al., 2007).
However, we believe that the reviewer is right and the need to incorporate students' perspectives should be included in the manuscript. We have included the following text in the conclusions section:
Future lines of research should include the perspective of other participants, such as students and parents, regarding PA. In addition, it is important to examine the barriers for practicing PA in different contexts (school environment, neighborhood), and identify any obstacles for the practice of PA among immigrant students of different nationalities, also from a gender perspective.
References:
- Creswell JW, Poth CN. Qualitative inquiry and research design. Choosing among five approaches. 4 ed. Thousand Oaks: SAGE, 2018.
- O'Brien BC, Harris IB, Beckman TJ, Reed DA, Cook DA. Standards for reporting qualitative research: a synthesis of recommendations. Acad Med. 2014;89(9):1245-1251.
- Tong A, Sainsbury P, Craig J. Consolidated criteria for reporting qualitative research (COREQ): a 32-item checklist for interviews and focus groups. Int J Qual Health Care. 2007;19(6):349–57.
In addition, it would be important to add that implications of the study are derived for teaching practice if the barriers to the practice of physical activity had actually been analyzed.
Response: We agree with the reviewer. We have edited the conclusions section inserting the following new text:
Concerning the implications for teaching, these findings may help PE teachers to identify elements such as family issues, beliefs, and economical issues that can condition PA. Furthermore, these results can help young or inexperienced teachers to plan and program the teaching of PE towards the needs of immigrant pupils by promoting greater participation, considering their possible limitations and perspectives. Also, these findings may be considered for the design and implementation of PA programs in certain social contexts. It is important for PA programs to: a) analyze the academic curriculum and devote increased time to PA, b) provide financial support to families who have limited resources in order to cover any associated costs related to children’s sports practice, and c) promote the organization of sports activities and multicultural games in community settings.
Definitely:
1.- Adapt the title of the article to the objective. Physical activity barriers are not really studied, but the teachers' perception of these barriers.
Response: Thank you for pointing this out. We have followed the reviewer´s comments. The title has been reworded, as follows:
The perspective of physical education teachers in Spain regarding barriers to the practice of physical activity among immigrant children and adolescents: A qualitative study.
2.- The introduction should describe the background on the research on barriers to physical activity in immigrants
Response: We have followed the reviewer´s comments. We have included further information regarding barriers to physical activity in immigrants.
We have included this information in the introduction section:
In Spain, the practice of PA in the school environment has been studied from the perspective of children [14], parents [15-17] and teachers [10, 15, 18, 19], both in the general population [14, 17] as well as specifically referring to the immigrant population [10, 15, 16, 18, 19]. In the study by Martinez-Andrés et al. [14] on the perception of the barriers encountered for the practice of PA in children between 8 and 11 years of age, the opportunities for the practice of PA are determined by the schedules established by their parents, in which the difficulty of reconciling work and family life and the prioritization of other types of academic activities over the practice of PA, together with the perceived danger of the community environment, favor more sedentary habits among the children. These results are consistent with studies carried out from the parents' perspective [15-17], in addition to identifying a greater difficulty for their children to participate in certain physical activities due to economic barriers [15-17], a greater difficulty of reconciling single-parent families, restriction of PA among children in informal contexts as a form of punishment, the lack of communication between the school and the family, the influence of the mass media [17], the inadequacy of infrastructure and lack of natural environments for practice [15-17], excessive homework [15, 16] and gender stereotypes and gender-related choices [10, 15-17, 20]. Moreover, Tamura et al. [21] identified that social environments of violence and insecurity in the neighborhoods where children and adolescents live are associated with a decrease in the practice of PA, representing the main structural barrier. These same authors recommend improving the spaces in which PA is carried out, such as parks and other open-air venues. Previous studies [22,23] point to the presence of crime in the neighborhood, the precariousness of the facilities where PA is performed [22,23], and high-traffic levels [23] as barriers to performing PA in children and adolescents in immigrant settings. Recently, Hu et al. [24], in their systematic review on factors influencing the performance of PA in children and adolescents, pointed to unsafe neighborhoods and inaccessibility of PA venues as barriers.
Moreover, we have included new references:
- Hu, D.; Zhou, S.; Crowley-McHattan, Z.J.; Liu, Z. Factors That Influence Participation in Physical Activity in School-Aged Children and Adolescents: A Systematic Review from the Social Ecological Model Perspective. Int. J. Environ. Res. Public Health 2021, 18, 3147. https://doi.org/10.3390/ijerph18063147
- Huang JH, Hipp JA, Marquet O, Alberico C, Fry D, Mazak E, Lovasi GS, Robinson WR, Floyd MF. Neighborhood characteristics associated with park use and park-based physical activity among children in low-income diverse neighborhoods in New York City. Prev Med. 2020 Feb;131:105948. doi: 10.1016/j.ypmed.2019.105948.
- Ross SE, Francis LA. Physical activity perceptions, context, barriers, and facilitators from a Hispanic child's perspective. Int J Qual Stud Health Well-being. 2016 Aug 16;11:31949. doi: 10.3402/qhw.v11.31949.
- Tamura K, Langerman SD, Orstad SL, Neally SJ, Andrews MR, Ceasar JN, Sims M, Lee JE, Powell-Wiley TM. Physical activity-mediated associations between perceived neighborhood social environment and depressive symptoms among Jackson Heart Study participants. Int J Behav Nutr Phys Act. 2020 Jul 10;17(1):91. doi: 10.1186/s12966-020-00991-y.
3.- Write the objective in a manner consistent with the title
Response: We have followed the reviewer´s suggestion, the objective has been rewritten as follows:
The aims were: a) to describe the perspectives of PE teachers on PA and b) barriers perceived, and c) to analyze the word frequency from interviews, to assess emotions and d) to describe the polarity of their perspective regarding PA.
4.- Increase the sample size or triangulate the population with interviews with families and students.
Response: We have followed the reviewer´s suggestion.
Regarding increasing the sample size, we have responded to this suggestion in a previous comment.
Regarding triangulating the population, we have responded to this suggestion in a previous comment.
5.- Detail the quantitative data by which the results are reached in each of the categories.
Response: We have followed the reviewer´s suggestion. Regarding the identification of categories from thematic analysis, we have answered this point in a previous comment.
In relation to the coding process of the thematic analysis, more information has been incorporated and a new figure has been added.
The researchers in this qualitative study conducted two types of analysis. On the one hand, an inductive thematic analysis, which focuses on identifying relevant fragments of the text, based on the relevant information they provide to answer the research question and its objectives. Thematic analysis occurs through the coding process, where the most relevant categories are identified, based on the information they provide (quality), and by consensus among researchers, not their frequency (Vaismoradi et al., 2014).
Furthermore, a second analysis of the qualitative material obtained from the interviews was carried out, a content analysis (Vaismoradi et al., 2014), based on the quantification of the most frequent words and through the use of dictionaries of standardised terms to obtain the polarity of the discourse/narrative of the teachers and the emotions in their discourse. This analysis has involved statistical treatment of the data.
Derived from these two different analysis processes, we obtain two types of results. The thematic results that appear in the section:
3.1. Results of the thematic analysis.
The results of the content analysis is presented in the following sections:
3.2. Results of word frequency (wordcloud)
3.3. Results of emotions and the acceptance-rejection (polarity) analysis
References:
- Vaismoradi M, Turunen H, Bondas T. Content analysis and thematic analysis: Implications for conducting a qualitative descriptive study. Nurs Health Sci. 2013 Sep;15(3):398-405. doi: 10.1111/nhs.12048. Epub 2013 Mar 11. PMID: 23480423.
6.- The results must be the result of the analysis of the interviews and not of the authors' conjectures
Response: We have followed the reviewer´s suggestions. Regarding the results obtained from interviews and not from the authors' conjectures, we have answered this point in previous comments.
7.- Describe the educational implications of these findings
Response: We have followed this suggestion. This point has been answered in a previous question.

Reviewer 2 Report
This is an interesting study about the subjective opinion of Spanish Physical Education Teachers on physical activity relationship of immigrant children and adolescents.
Here are some comments about this manuscript.
Results
Line 209: Eliminate full stop.
Figure 1 is not cited on the text.
Results section lacks on important information of professional experience of PE teachers, their age, gender, cultural beliefs, since these variables could influence their opinions. Also, results lack on comparing children and adolescents towards PE lessons, as it is well-known that there are differences due to adolescents' entrance in puberty and changes in body image and body composition.
Discussion
The manuscript lacks on discussing about differences between children and adolescents towards PE lessons, and differences in the published literature in relation to PE teachers’ opinions and parents’.
Limitations of the study should include that participants were teachers and considering the religious beliefs as a possible barrier to the practice of PA may be a subjective opinion of them, and not the real parents’ opinion or believes. In addition, authors indicate the study has several limitations; however, only 2 are indicated and the first is not explained.
Therefore, considering the small size of the sample, authors should explain why their results are original and the reason for that small number of participants.
In addition, since physical education is not considered as important as expected for parents, why only PE teachers were interviewed and parents were not.
Conclusions
The identified barriers for performing PA among immigrant children should be indicated.
Author Response
ijerph-1203384
Title: Barriers to the practice of physical activity among immigrant children and youth: A qualitative study on the experiences of physical education teachers in Spain.
We would like to thank the Editors and the Reviewers for their careful consideration of our manuscript. We would also like to thank the Reviewers’ suggestions, which we believe have enhanced the quality of the manuscript. We have highlighted all the changes we have made throughout the text in green highlight. Below, please find a detailed list of how we have addressed each comment.
In addition, the authors include comments regarding the duplicate report at the end of the response letter. This information is provided for the publisher / publishers.
Sincerely,
The Authors.
REVIEWER 2
Comments and Suggestions for Authors
This is an interesting study about the subjective opinion of Spanish Physical Education Teachers on physical activity relationship of immigrant children and adolescents.
Here are some comments about this manuscript.
Results
Line 209: Eliminate full stop.
Response: Eliminated as suggested.
Figure 1 is not cited on the text.
Response: Thanks for pointing this out, this is now corrected.
Results section lacks on important information of professional experience of PE teachers, their age, gender, cultural beliefs, since these variables could influence their opinions.
Response: We have included further information regarding the PE teachers.
The study began in September 2019 and ended in September 2020. Twenty participants were included, with a mean age of 40.9 years (SD 6.6) and with 15.55 years (SD 9.2) of PE teaching experience. Of the 20 participants, 13 were male and 7 were female. All participants were currently working in the southern area of the Community of Madrid (Spain). Of the participants included, 8 did not believe in any religion, whereas 12 considered themselves to be Catholic Christians. Of the latter, 6 defined themselves as practising Catholics.
Also, results lack on comparing children and adolescents towards PE lessons, as it is well-known that there are differences due to adolescents' entrance in puberty and changes in body image and body composition.
Response: We agree that the process of teaching physical education is different between children and adolescents, however, the teachers have not provided narratives and/or information regarding the differences in the process of teaching the subject of physical education between children and adolescents. They have provided information on the barriers they perceived during the promotion of physical activity. Unfortunately, they have not addressed this aspect specifically. Therefore, we believe that this information needs to be included in the limitations.
The following text has been added to the limitations section:
Also, teachers were not asked about the process of teaching PE to children and adolescents, which may reveal differences in the performance of PA.
Discussion
The manuscript lacks on discussing about differences between children and adolescents towards PE lessons, and differences in the published literature in relation to PE teachers’ opinions and parents’.
Response. We have included two new paragraphs which describe differences between children and adolescents towards PE lessons, and differences in the published literature in relation to the opinions of PE teachers and that of parents in the discussion section.
Regarding differences between children and adolescents towards PE lessons.
Additionally, there are differences between children and adolescents concerning their attitude towards PE lessons. Previous studies report that interest towards PA practice and PE decreases as students get older [54-56]. Gómez Rijo et al. [56] describe how the motivation of adolescents in secondary education towards PE practice is lower than that of primary school students. According to Fraile Aranda & Catalina Sancho [57], primary school students tend to feel more skillful at a sporting level than secondary school students. Overall, secondary students have a lower self-concept regarding their ability to perform PA compared to younger students [57].
Among teachers, there are also differences in their perception and orientation towards PA in primary and secondary education influenced by the school curriculum [58-59]. Primary PE teachers orientate learning towards attitudinal and self-realization goals, while secondary school teachers give more importance to procedural and conceptual learning, developing a greater social responsibility [58]. As a result, primary school teachers are more oriented towards games and global developmental work, while secondary school teachers are oriented towards more social skills development. However, Sicilia Camacho et al. [59] point out that in primary education, priority should be given to the development of social relations whereas in secondary education the focus should be on the acquisition of healthy habits.
We have included new references:
- Fraile Aranda, A.; Catalina Sancho, J. Diferencias en autoconcepto físico en escolares de primaria y secundaria. Lúdica pedagógica. 2013, 2(18), 93-102.
- Gómez Rijo, A.; Gámez Medina, S.; Martínez Herráez, I. Efecto del género y la etapa educativa del estudiante sobre la satisfacción y la desmotivación en educación física durante la educación obligatoria. Ágora para la EF y el deporte. 2011, 13(2), 183-196.
- Moreno, A. y Hellín, P. ¿Es importante la Educación Física?. Su valoración según la edad del alumno y el tipo de centro. Revista Internacional de Medicina y Ciencias de la Actividad Física y el Deporte. 2004, 8.
- Moreno, J. A.; Rodríguez, P. L. y Gutiérrez, M. Intereses y actitudes hacia la Educación Física. Revista Española de Educación Física. 2003, XI, 2, 14-28.
- Sicilia Camacho, A.; Sáenz-López Buñuel, J.; Manzano Moreno, I.; Delgado Noguera, M. A.; El desarrollo curricular de la Educación Física en Primaria y Secundaria: un análisis desde la perspectiva del profesorado. Apunts: Educación física y deportes. 2009, 98, 23-32.
- Sicilia, A.; Delgado, M. A.; Sáenz-López, P.; Manzano, J. I.; Varela, R.; Cañadas, J. F. & Gutiérrez, M. La evaluación de aprendizajes en educación física. Diferencias en función del nivel educativo. Motricidad European Journal of Human Movement. 2006, 17, 71-95.
Regarding differences in the published literature in relation to the opinion of PE teachers and parents, the following new text has been added.
Regarding differences in teachers’ and parents’ perspectives on PE, Porcuna & Rodríguez-Martín [67] reported that both parents and teachers agree that they can act as role models for children and adolescents, which can positively and negatively influence them to practice PA. Similarly, they agree that there are environmental barriers to PA, such as high workloads, lack of parental time, or the lack of spaces where PA can be carried out [67]. However, in the same study, Porcuna & Rodríguez-Martín [67] described that while teachers demanded greater collaboration between parents and teachers, parents demanded to receive more information from teachers on how to promote PA outside the school context [67]. Similarly, from the parents' perspective, sex and gender are identified as determining factors in the practice of PA [10, 15-17, 20], along with economic barriers and lack of infrastructure for PA [15-17].
We have included new references:
- Porcuna VA, Rodríguez-Martín B. Parents' and Teachers' Perceptions of Physical Activity in Schools: A Meta-Ethnography. J Sch Nurs. 2020 Nov 26:1059840520972005. doi: 10.1177/1059840520972005. Epub ahead of print. PMID: 33243055.
Limitations of the study should include that participants were teachers and considering the religious beliefs as a possible barrier to the practice of PA may be a subjective opinion of them, and not the real parents’ opinion or believes. In addition, authors indicate the study has several limitations; however, only 2 are indicated and the first is not explained.
The authors do not agree that the teachers' perspective is a limitation. In this study, the description of their perspective is the object of study. In qualitative research, the focus is on the description and analysis of the perspective of groups of people regarding concrete phenomena.
To determine the objective of the study, this was based on the following study question: “The questions guiding this study were: How do physical education teachers perceive PA in immigrant children and adolescents?” The research question directing the study already includes the fact that the aim of this study will be to describe the teachers' perspective on physical activity (phenomenon under study). This way of constructing the question and directing the research objective is used in qualitative research.
We do not see the fact that we describe the perspective of a group of people as a negative aspect or a limitation, but rather as part of the design of qualitative research studies aimed at describing and analysing the perspective of groups living in or in contact with certain phenomena.
Carpenter & Suto (2008) reported that: “…qualitative research is well suited to answer questions that ask what particular experiences are like and how people create meaning from their circumstances. These types of questions are appropriate for topics where Little is known from the client´s point of view, and much depends on how knowledge is created…” (page 21).
However, we agree with the reviewer that information needs to be included to clarify that teachers' perspective, which does not mean that it coincides with or is contrary to the perspective of other participants such as parents.
For this reason, a first clarifying paragraph has been added before the limitations:
The results of this study reveal the perspective of PE teachers on aspects that may condition the practice of PA in immigrant children and adolescents. This perspective is the object of the study, however, it does not mean that it represents, coincides with or is contrary to the perspective or beliefs of other participants involved in the practice of PA, such as parents and students. This study has several limitations. First, a small number of participants was included. The present study has included 20 participants. Previous qualitative studies [25,30] describe how the total number of participants included does not depend on a previous calculation of the sample size, rather it is based on the saturation or redundancy of the information obtained in the interviews. Turner-Bowker et al. [68] reported that 92-97% of the saturation can be analyzed after interview number 15 and 20. Secondly, the teachers included in the study were not part of the liaison classrooms, classrooms intended for the support and adaptation of immigrant children/adolescents joining the school. This could influence the results. Also, teachers were not asked about the process of teaching PE to children and adolescents, which may reveal differences in the performance of PA. Finally, the qualitative nature of this study, gathering the perspective and experiences of a group of PE teachers concerning a certain phenomenon, cannot be generalized. However, it allows us to gain a deeper understanding of the phenomenon from their perspective.
Therefore, considering the small size of the sample, authors should explain why their results are original and the reason for that small number of participants.
Response: We thank the reviewer for the first comments, however, regarding the last observation, the authors de not agree with the statement that the small number of patients of the same undermine the value of this study.
Regarding the number of patients, the reasons are as follows; in qualitative research there is not a number of participants that one can calculate previously (Creswell & Poth, 2018; Carpenter & Suto, 2008), in the same manner, in qualitative research we do not seek to extrapolate the results to the general population (Creswell & Poth, 2018; Pope & Mays, 2006), rather, the attempt is to deepen our knowledge of the complex phenomena (physical activity promotion) or impact of programs and interventions on groups of people who live specific situations.
“The intent in qualitative research is not to generalize the information but to elucidate the particular, the specific.” (Creswell & Poth, 2018.p.158).
“In quantitative studies, power calculations determines which sample size (N) is necessary to demonstrate effects of a certain magnitude from an intervention. For qualitative interview studies, no similar standards for assessment of sample size exist.” (Malterud et al., 2015.p.1)
“Qualitative researchers actively seek to recruit participants who can represent well and have experience of the phenomenon of interest (…)The key is to focus the participant selection strategically and meaningfully, rather than to create a representative sample (…) The sample size and range of participants are intended to generate sufficient data to explore processes, similarities and differences, to develop theory and descriptions that take into account specific contexts, rather than to make statistical comparisons or infer causal relationships. In qualitative research there is no set formula for determining the sample size (…) The quality and comprehensiveness of the data acquired varies with different participants or sources of information. Some participants have more experiences with the topic of interest, are able to reflect deeply on their experiences (…) The greater the quality of data acquired from each source, the smaller the sample size required.” (Carpenter & Suto, 2008.p.80).
Besides, the inclusion of a greater or lesser number of participants depends on the richness of the information that is obtained and the presence of repetition of this information among the participants who are included (Creswell & Poth, 2018; Carpenter & Suto, 2008; Malterud et al., 2015). In qualitative research, this concept is known as “data saturation” or “information redundancy” (Malterud et al., 2015). In this manner, in this study, 20 participants were included because in this number data redundancy was obtained (saturation) (Creswell & Poth, 2018; Carpenter & Suto, 2008; Malterud et al., 2015) this means that the contents of the interviews analyzed present repetitive narratives or results, therefore it is not necessary to continue with the data collection and/or the inclusion of more participants (Creswell & Poth, 2018; Carpenter & Suto, 2008). Recently, Turner-Bowker et al. (2018), reported that saturation or redundant information is described as the point at which no new data is expected to emerge from the performance of additional qualitative interviews. These authors in their studies showed to assess the point at which saturation of concept was achieved in a retrospective evaluation of 26 interview studies between 2006 and 2013. The findings reported that 92% of redundant information emerged by the 15th interview and 97% emerged by the 20th interview. For this reason, the authors of this study believe that are number of participants (n=20) is optimal and well suited to the saturation of data obtained after the analysis of our interviews following the qualitative methodology and the recommendations for qualitative studies in health sciences established by the Consolidated criteria for reporting qualitative research (COREQ) (Tong et al., 2007) and the Standards for reporting qualitative research (SRQR) (O'Brien et al., 2014).
However, following the reviewer’s recommendations, we believe it is necessary to include more information in the manuscript (limitations section) so that this is clearer.
We have included this text in the ‘limitations’ section:
This study has several limitations. First, a small number of participants was included. The present study has included 20 participants. Previous qualitative studies [25,30] describe how the total number of participants included does not depend on a previous calculation of the sample size, rather it is based on the saturation or redundancy of the information obtained in the interviews. Turner-Bowker et al. [68] reported that 92-97% of the saturation can be analyzed after interview number 15 and 20.
References:
- Carpenter C, Suto M. Qualitative research for occupational and physical therapists: A practical guide. Oxford: Black-Well Publishing, 2008.
- Creswell JW, Poth CN. Qualitative inquiry and research design. Choosing among five approaches. 4 ed. Thousand Oaks: SAGE, 2018.
- Malterud K, Siersma VD, Guassora AD. Sample Size in Qualitative Interview Studies: Guided by Information Power. Qual Health Res. 2015 Nov 27. DOI: 10.1177/1049732315617444
- O'Brien BC, Harris IB, Beckman TJ, Reed DA, Cook DA. Standards for reporting qualitative research: a synthesis of recommendations. Acad Med. 2014;89(9):1245-1251.
- Pope C, Mays N. Qualitative research in health care. Oxford: BMJ Books & Blackwell Publishing; 2006.
- Tong A, Sainsbury P, Craig J. Consolidated criteria for reporting qualitative research (COREQ): a 32-item checklist for interviews and focus groups. Int J Qual Health Care. 2007;19(6):349–57.
- Turner-Bowker DM, Lamoureux RE, Stokes J, Litcher-Kelly L, Galipeau N, Yaworsky A, Solomon J, Shields AL. Informing a priori Sample Size Estimation in Qualitative Concept Elicitation Interview Studies for Clinical Outcome Assessment Instrument Development. Value Health. 2018 Jul;21(7):839-842. doi: 10.1016/j.jval.2017.11.014.
We have included a new reference to the manuscript:
- Turner-Bowker DM, Lamoureux RE, Stokes J, Litcher-Kelly L, Galipeau N, Yaworsky A, et al. Informing a priori Sample Size Estimation in Qualitative Concept Elicitation Interview Studies for Clinical Outcome Assessment Instrument Development. Value Health 2018;21:839–42. PubMed https://doi.org/10.1016/j.jval.2017.11.014
In addition, since physical education is not considered as important as expected for parents, why only PE teachers were interviewed and parents were not.
Response:
The research question of this study was to understand the perspective of the teachers, not the parents. Therefore, following the research question, the study was constructed using a method and design that answered the question; that of qualitative research. Focusing on those participants who could answer that question, in this case the teachers. If the question where to be changed, the research question could be changed or new participants could be included, but that would mean conducting a different study.
By obtaining results that showed a perspective that coincided with or was contrary to other groups (parents) not included in the study, it does not mean that these aspects were not addressed in the present study. In fact, observations are included in the discussion.
The authors believe that it would not be possible to include other participants, because the study had already been completed. If we wanted to include the parents' perspective or the combined perspective, that means starting a new study, with a new research question, with a new research protocol, where new participants are included, and having to resubmit the new study protocol to the research ethics committee.
Conclusions. The identified barriers for performing PA among immigrant children should be indicated.
Response: We included the following new text:
Our findings help further our understanding of the barriers related with PA in immigrant children and youth in Spain such as giving more importance to certain subjects because of their job prospects, having low economic resources, the lack of role models in the family for the practice of PA, the formation of closed groups for PA practice among students, and the limitation of PA in girls due to gender differences. One of the strengths of this study is that it describes the barriers for performing PA among immigrant children from the perspective of the PE teachers.
We hope that this revision is satisfactory and that the manuscript is now suitable for publication in IJERPH.
Sincerely,
The Authors

Round 2
Reviewer 1 Report
The authors have made a great effort to answer all the questions raised, justifying each of the decisions taken.